# The Quest for a Respiratory Syncytial Virus Vaccine for Older Adults: Thinking beyond the F Protein

**DOI:** 10.3390/vaccines11020382

**Published:** 2023-02-07

**Authors:** Victoria A. Jenkins, Bernard Hoet, Hubertus Hochrein, Laurence De Moerlooze

**Affiliations:** 1Bavarian Nordic AG, 6301 Zug, Switzerland; 2Bavarian Nordic GmbH, 82152 Munich, Germany

**Keywords:** immunity, older adults, vaccine efficacy, RSV, vaccination

## Abstract

Respiratory syncytial virus (RSV) is a common cause of paediatric respiratory tract infection and causes a significant health burden in older adults. Natural immunity to RSV is incomplete, permitting recurrent symptomatic infection over an individual’s lifespan. When combined with immunosenescence, this increases older adults’ susceptibility to more severe disease symptoms. As RSV prophylaxis is currently limited to infants, older adults represent an important target population for RSV vaccine development. The relationship between RSV and our immune systems is complex, and these interactions require deeper understanding to tailor an effective vaccine candidate towards older adults. To date, vaccine candidates targeting RSV antigens, including pre-F, F, G (A), G (B), M2-1, and N, have shown efficacy against RSV infection in older adults in clinical trial settings. Although vaccine candidates have demonstrated robust neutralising IgG and cellular responses, it is important that research continues to investigate the RSV immune response in order to further understand how the choice of antigenic target site may impact vaccine effectiveness. In this article, we discuss the Phase 3 vaccine candidates being tested in older adults and review the hurdles that must be overcome to achieve effective protection against RSV.

## 1. Introduction

Respiratory syncytial virus (RSV) poses a significant burden to global health. It is a common cause of paediatric respiratory tract infection and is a health risk to older adults, with similar morbidity and mortality rates to influenza [1]. In the United Kingdom, RSV-attributable respiratory disease is estimated to cause 83 paediatric deaths per average season [2]; this rises to 8482 deaths per year among adults aged ≥18 years, with 93% of those cases occurring in older adults aged ≥65 years [3]. A recent meta-analysis estimated that 5.2 million cases of RSV occurred in high-income countries among adults aged ≥60 years in 2019, leading to 470,000 hospitalisations and 33,000 in-hospital deaths [4]. Limited epidemiological data are available for developing countries [4,5]. These figures are likely to be underestimated as they only consider those admitted to hospital, omitting those without access to care and those with suboptimal healthcare-seeking behaviour [5].

Natural RSV infection results in incomplete immunity, permitting recurrent symptomatic infection over an individual’s lifespan [6,7]. RSV infects nearly all children by the time they reach 2 years of age, and those aged under 6 months are at higher risk of severe RSV infection as the immune system is immature at this stage [8]. In older adults, underlying comorbidities and age-related progressive decline in immune function—immunosenescence—increase the risk of severe outcomes following RSV infection [9]. The exact mechanism responsible for increased RSV severity in older adults remains to be elucidated.

RSV prophylaxis is currently limited to passive immunisation with monoclonal antibodies: palivizumab is approved for infants at high risk for RSV disease and nirsevimab gained approval for all infants in November 2022 by the European Medicines Agency (EMA) [10,11,12,13]. Despite substantial research efforts over four decades, there are no licensed RSV vaccines available for infants or older adults. This does not mean that the quest for an RSV vaccine is over; multiple candidates are in late-stage clinical trials (or have reported preliminary data and are undergoing regulatory file submission) using various RSV antigen targets. Here, we review the current literature on RSV and discuss important factors to consider for the effectiveness of an RSV vaccine, including promising vaccine candidates for older adults.

## 2. RSV Pathogenesis

RSV primarily infects and replicates within the epithelium of the upper respiratory tract, causing mild rhinorrhoea symptoms, coughing, or wheezing. Viral penetration of the epithelial mucus layer activates the innate immune response but this is not sufficient to clear the virus; the adaptive immune response must contribute to control the RSV infection [1]. CD4^+^ and CD8^+^ T cell responses are among the adaptive immune responses mediating viral clearance and correlating with protection against RSV re-infection after a primary RSV infection, as well as circulatory neutralising IgG and secretory IgA [7,14]. However, antibody titres are often inadequate for achieving protection against disease caused by RSV in most adults, despite a lifetime of RSV exposure [14]. This allows the virus to repeatedly re-infect a host, which ultimately ensures its success [14]. What underlies this incomplete immunity against RSV infection remains unclear, yet a lack of durable humoral and cellular responses at a magnitude needed to prevent infection is considered a probable cause [15]. A deeper comprehension of RSV immunology is fundamental to understanding vaccine design.

### 2.1. Virus Structure

RSV is a negative-sense, single-stranded RNA virus of the family *Pneumoviridae*. The RSV genome consists of 10 open reading frames that encode 11 structural and non-structural proteins (Figure 1) [1,12]. The viral envelope possesses three integral transmembrane proteins: the receptor attachment glycoprotein (G), the fusion glycoprotein (F), and the small hydrophobic (SH) protein. Other structural proteins include a nucleoprotein (N), a large RNA polymerase (L), a phosphoprotein (P), a matrix protein (M), and transcription factors (M2-1 and M2-2). The two non-structural (NS) proteins produced are NS1 and NS2 [1,16]. RSV is classified into two major antigenic subtypes: RSV A and RSV B, which may co-circulate during a given season but often one subtype dominates in 1–2-year intervals [1,17]. Whereas the F protein shows considerably high homology between RSV subtypes, the G proteins are antigenically more distant [1]. 

### 2.2. RSV F Antigen and Approved Prophylactic Therapies

The F protein is located on the surface of the virion and is the only highly conserved surface antigen across all known RSV subtypes [9,19]. It is essential for viral fusion and cell entry and produces strong RSV-neutralising antibodies [19]. Consequently, it is the target of most RSV vaccine candidates [20]. The F protein exists in two conformations. The metastable prefusion conformation (pre-F) rapidly undergoes several structural changes, both spontaneously and following host cell binding, resulting in a secondary, more thermodynamically stable post-fusion (post-F) conformation [21]. The pre-F conformation possesses six major antigenic sites (Ø, I, II, III, IV, V), of which only I, II, III, and IV are present in the post-F conformation [22].

Palivizumab is a monoclonal antibody targeting antigenic site II, which is present on both the pre- and post-F conformations. Palivizumab is indicated by the EMA and the US Food and Drug Administration (FDA) and administered to infants at high risk for RSV in five doses—the first dose is recommended prior to the commencement of the RSV season, with subsequent doses administered monthly throughout the season [10,11,23]. A test-negative case-control trial of 849 high-risk children younger than 24 months was conducted to determine the effectiveness of palivizumab (administered at 15 mg/kg). When administered monthly during the RSV season, palivizumab was 58% (95% confidence interval [CI], 43.1–69) effective in the prevention of RSV-related hospitalisations after adjusting for underlying differences and 62.1% (95% CI, 35.1–77.9) effective in the prevention of RSV-related intensive care admissions [24]. The prophylactic efficacy of palivizumab was investigated in a double-blind, exploratory study of 49 healthy adults inoculated with RSV administered at 8 or 15 mg/kg; however, no significant difference in RSV viral load was observed 2–12 days post-viral challenge between the placebo and palivizumab groups (11.68 vs. 9.94 Log10 copies*day/mL; *p* = 0.5802) [25]. 

Nirsevimab is a monoclonal antibody targeting the antigenic site Ø, which is only expressed on the pre-F conformation. Nirsevimab was granted EMA and FDA regulatory designations to facilitate expedited development for all infants as a single dose [12]. The EMA’s Committee for Medicinal Products for Human Use adopted a positive opinion for the approval of nirsevimab, which, along with the results of the clinical development programme, led to approval by the EMA [13,26,27]. Evaluation of a single dose (50 mg for a weight of <5 kg or 100 mg for a weight of ≥5 kg) of nirsevimab in healthy late-preterm and term infants entering their first RSV season resulted in a 74.5% (95% CI, 49.6–87.1) lower incidence of medically attended RSV-associated lower respiratory tract infection versus the placebo, which was consistent across RSV subtypes [28]. Nirsevimab was 62.1% (95% CI, –8.6 to 86.8) effective in the prevention of hospitalisation for RSV-associated lower respiratory tract infection [28]. The neutralising potency of nirsevimab was up to 100-fold greater than palivizumab in vitro and it had a longer serum half-life [28,29]; these data may suggest the induction of more effective antibody-mediated protection with preferential targeting of pre-F-specific site Ø [19,21,22]. Although both monoclonal antibodies demonstrate protective efficacy in infants, the 58–62% prevention rates against hospitalisation in infants [24,28] and lack of prophylactic efficacy in adults with palivizumab [25] show the need to improve our understanding of immune correlates of protection against RSV. 

## 3. Considerations for an Effective RSV Vaccine in Older Adults: RSV Target Antigens and the Immune Response

One of the most recognised consequences of ageing is a decline in immune function. This is characterised by a reduction in peripheral naïve T cell numbers and an accumulation of memory T cells, coupled with an increase in CD8^+^ T cell numbers and a decrease in CD4^+^ T cell numbers. Likewise, IgD^−^ and IgM-producing naïve B cell populations are replaced by IgG^−^ and IgA-producing memory B cells, accompanied by impaired production of high-affinity antibodies [30,31]. These immune alterations, together with chronic low-grade inflammation, diminish responses to newly encountered antigens and are considered the hallmarks of immunosenescence [30,31]. Understanding the full spectrum of changes that characterise immunosenescence is fundamental to the development of novel and improved vaccines to target the many infectious diseases responsible for substantial morbidity in this population [32,33]. For optimal RSV-specific immune responses in older adults, multiple strategies may need to be considered, including a diverse induction of innate immune responses to employ multiple protective adaptive immune responses against various potent RSV target antigens [9].

### 3.1. The F Protein

A natural infection study in adults showed that over 60% of the most potent RSV-neutralising antibodies target antigenic sites Ø and V, which are only present on the pre-F conformation. Meanwhile, antibodies targeting sites I–IV showed a wide range of neutralising potencies, including moderate to non-neutralising [34]. Vaccine studies targeting the post-F RSV protein alone, however, have failed in their efficacy trials. A Phase 2b study evaluated RSV post-F protein with a glucopyranosyl lipid A-based adjuvant (a synthetic analogue of monophosphoryl lipid A, in a 2% stable emulsion) in subjects aged ≥60 years [35]. The vaccine candidate failed to demonstrate vaccine efficacy; the incidence of RSV-associated acute respiratory tract infection after vaccination was 1.7% in the RSV vaccine arm and 1.6% in the placebo arm, for an estimated vaccine efficacy of −7.1% (90% CI, −106.9% to 44.3%) [35]. The Phase 3 Resolve™ trial evaluated a post-F nanoparticle vaccine in 11,856 older adults aged ≥60 years. Similarly, the pre-specified primary and secondary efficacy objectives were not met and the post-F vaccine candidate did not demonstrate vaccine efficacy [36]. The preference for highly potent antibodies to target pre-F-specific sites over post-F sites may explain the failure of vaccines targeting the post-F protein to elicit robust protective efficacy [9]. The greater neutralising potency of pre-F-targeting nirsevimab compared with palivizumab adds weight to this idea, as discussed earlier in Section 2.2 [23,29]. 

Five RSV vaccine candidates are in the end phase of development for older adults: four targeting only the pre-F protein (RSVpreF, GSK3844766A, Ad26.RSV.preF + preF subunit, and mRNA-1345) and one targeting the F, G (A), G (B), M2-1, and N proteins (MVA-BN-RSV; Table 1). RSV vaccine candidates targeting the pre-F protein have demonstrated robust neutralising responses in healthy and older adults in Phase 1 and 2 clinical trials, which have paved the way for Phase 3 testing. RSV-neutralising antibody titres and IgG levels generally peak in the month after the first vaccine dose and decline over time, yet they remain above baseline at 12 months [37,38,39,40,41]. This decline over time raises questions around the durability of the humoral response, which may contribute to incomplete RSV immunity [15]. However, a second dose administered 12 months later can elicit a booster response, increasing neutralising titres to levels comparable to peak responses [41]. Three candidates have announced positive top-line data from pre-planned interim analyses of their Phase 3 clinical trials. GSK3844766A contains a recombinant subunit pre-F antigen (RSVPreF3) and AS01_E_ adjuvant and has been granted FDA priority review designation [42]. It demonstrated vaccine efficacy of 82.6% (96.95 CI, 57.9–94.1) and 94.1% (95% CI, 62.4–99.9) against RSV-associated lower respiratory tract disease and severe RSV lower respiratory tract disease, respectively, in older adults aged ≥60 years. Vaccine efficacy against lower respiratory tract disease was consistent across RSV A and B subtypes, with no unexpected safety concerns [43,44,45]. Meanwhile, RSVpreF, a bivalent subunit vaccine that received FDA breakthrough designation following positive Phase 2a human challenge trial data [46], demonstrated vaccine efficacy of 66.7% (96.66% CI, 28.8–85.8) and 85.7% (96.66% CI, 32.0–98.7) for protection against RSV-associated lower respiratory tract illness with ≥2 symptoms and ≥3 symptoms, respectively. RSVpreF was well tolerated with no safety concerns [47,48]. Additionally, mRNA-1345, an mRNA-based vaccine encoding for stabilised pre-F, demonstrated vaccine efficacy of 83.7% (95.88% CI, 66.1–92.2) and 82.4% (96.36% CI, 34.8–95.3) for protection against RSV-associated lower respiratory tract illness with ≥2 symptoms and ≥3 symptoms, respectively. mRNA-1345 was well tolerated with no safety concerns identified. Based on these interim results, Moderna announced its intention to submit mRNA-1345 for regulatory approval [49].

There are potential risks when depending on the activity of neutralising antibodies against a few specific epitopes, such as Ø of the RSV pre-F protein. Targeting a single epitope on the F protein could carry a risk of antibody-escape mutant viruses and thus require ongoing molecular surveillance to ensure continued neutralisation activity [50]. For instance, suptavumab, a monoclonal antibody targeting the RSV pre-F protein antigenic site V, was 10- and 5-fold more potent than palivizumab in neutralising RSV A and B, respectively. Yet in a Phase 3 trial in infants, suptavumab failed to meet its primary endpoint of reducing hospitalisation or lower respiratory tract infection. This was attributed to suptavumab epitope mutations found at site V on circulating RSV B strains, thereby rendering it ineffective [50]. As previously mentioned, robust immune responses and consistently high vaccine efficacy against the RSV A and B subtypes have been observed with GSK3844766A [44,45]. However, as analyses were conducted following one RSV season, the vaccine’s long-term efficacy for ensuring cross-reactivity across multiple RSV seasons on a global scale of mass vaccination is unknown. Even though mutation rates are higher for severe acute respiratory syndrome coronavirus 2 (SARS-CoV-2) (another single-stranded RNA virus) compared with RSV [51,52], ongoing surveillance of vaccination against SARS-CoV-2 has revealed the emergence of variants harbouring mutations to the main spike protein target of neutralising antibodies [53]. While many SARS-CoV-2 variants continue to be potently neutralised following mass vaccination, some can escape vaccine-induced humoral immunity [53]. Regarding RSV, widespread use of a vaccine that induces F-specific antibodies could promote a selective advantage for RSV mutants with amino acid changes in the relevant F protein epitopes [54]. A comprehensive analysis of sequence variations between the F proteins of circulating RSV A and RSV B demonstrated a high degree of sequence conservation in adults. However, preferential accumulation of amino acid changes at certain antigenic sites was noted, particularly for RSV B at antigenic sites Ø, V, and I [54]. Neutralising responses predominantly target pre-F sites Ø and V, indicating that alterations at either site could have a substantial impact on the ability of a pre-F-only vaccine to provide durable protection in adults [54]. With the potential for selective pressure to escape mutants, it is essential that vaccine candidates are closely monitored long term to determine the persistence of cross-reactivity.

**Table 1 vaccines-11-00382-t001:** Neutralising antibody responses to current Phase 3 RSV vaccine candidates in healthy and older adults in clinical development.

Vaccine Candidate, Sponsor	Formulation, Antigen	Clinical Trial Phase	Populations Assessed	N	Neutralising Antibody * GMFRs at up to 1 Month vs. Baseline	Neutralising Antibody * GMFRs at ≥12 Months vs. Baseline	Vaccine Efficacy (VE) *
RSVpreF, Pfizer [37,38,55]	Bivalent subunit, Prefusion F	Phase 1/2	Adults aged 18–49 years	618	RSV A: 10.6–16.9 RSV B: 10.3–19.8	RSV A: 3.9–5.2 RSV B: 3.7–5.1	–
		Phase 1/2	Older adults aged 65–85 years	317	RSV A: 4.8–11.6 RSV B: 4.5–14.1	RSV A: 2.1–3.5 RSV B: 2.2–4.3	–
		Phase 2a human challenge trial	Adults aged 18–50 years	62	RSV A: 20.5 (95% CI, 16.6–25.3) RSV B: 20.3 (95% CI, 15.6–26.4)	–	VE against RT-PCR-confirmed symptomatic RSV infection with ≥1 symptom: ^†^ 86.7% (95% CI, 53.8–96.5) VE against quantitative culture-confirmed RSV infection with ≥1 symptom: ^‡^ 100.0% (95% CI, 67.7–100.0)
		Phase 3	Older adults aged ≥60 years	34,284 (n = 1050)	–	–	VE against RSV-associated LRTD: 66.7 (96.66% CI, 28.8–85.8%)–85.7% (96.66% CI, 32.0–98.7) VE against RSV-associated ARI: 62.1% (95% CI, 37.1–77.9)
GSK3844766A, GSK [39,44,45,56]	Recombinant subunit, Prefusion F + AS01_E_ adjuvant	Phase 1/2	Adults aged 18–40 years	48	RSV A: 7.5–13.7	–	–
		Phase 1/2	Older adults aged 60–80 years	1005	RSV A: 5.6–9.9	RSV A: 2.7–4.4	–
		Phase 3	Older adults aged ≥60 years	1653 (n = 566)	RSV A: 10.5 RSV B: 7.8	RSV A: ^§^ 4.4 RSV B: ^§^ 3.5	
		Phase 3	Older adults aged ≥60 years	26,664 (n = 24,960)	–	–	VE against RSV-associated LRTD: 82.6 (96.95% CI, 57.9–94.1)–94.1% (95% CI, 62.4–99.9) VE against RSV-associated ARI: 71.7% (95% CI, 56.2–82.3)
Ad26.RSV.preF, Janssen [41,47,57,58,59]	Adenoviral vector 26, Prefusion F	Phase 1	Older adults aged >60 years	72	RSV A and B: Increased compared with baseline ^¶^	Maintained above baseline levels 2 years after vaccination	–
		Phase 2a human challenge trial	Adults aged 18–50 years	63	RSV A: 5.8	–	VE against RT-PCR-confirmed RSV infection ≥1 symptom: ** 51.9% (95% CI, –7.4–83.2) VE against quantitative culture-confirmed RSV infection with ≥1 symptom: ^‡^ 55.1% (95% CI, 9.4–82.2)
	Adeno 26 viral vector Prefusion F + Prefusion F subunit	Phase 2b	Older adults aged ≥65 years	5782	RSV neutralising antibodies: ^††^ 13.5 RSV pre-F-specific antibodies: ^††^ 8.6	RSV neutralising antibodies: 2.8 RSV pre-F-specific antibodies: 2.1	VE against RSV-associated LRTD: up to 80% (95% CI, 52.2–92.9; *p* < 0.001) VE against RSV-associated ARI: 69.8% (95% CI, 42.7–85.1) VE against RT-PCR-confirmed RSV-associated LRTD at 3-year follow-up: 78.7% (95% CI, 57.3–90.4) VE against RT-PCR-confirmed RSV-associated ARI at 3-year follow-up: 65.7% (95% CI, 43.5–79.9)
mRNA-1345, Moderna [49,60]	mRNA, Prefusion F	Phase 1	Older adults aged 65–79 years ^‡‡^	n = 298	RSV A: 9.9–16.6 RSV B: 5.3–12.6	RSV A: ^††^ 3.1–5.8 RSV B: ^††^ 2.9–5.5	–
		Phase 3	Older adults aged >60 years	~37,000	–	–	VE against RSV-associated LRTD: 83.7% (95.88% CI, 66.1–92.2)–82.4% (96.36% CI, 34.8–95.3)
MVA-BN-RSV, Bavarian Nordic [61,62,63,64,65,66]	Bivalent vector-based, RSV F, G (A), G (B), M2-1 and N	Phase 1	Adults aged 18–49 years	63	1.2–1.9 ^§§,¶¶^	–	–
			Older adults aged 50–65 years		1.0–2.0 ^§§,¶¶^	–	–
		Phase 2	Older adults aged ≥55 years	420	1.3–3.4 ^††,§§^	Maintained above baseline levels 1 year after vaccination	–
		Phase 2a human challenge trial	Adults aged 18–50 years	61	RSV A: 2.2 ^††^ RSV B: 1.6 ^††^	RSV A: 1.7 ^§^ RSV B: 1.3 ^§^	VE against RT-PCR-confirmed RSV infection with ≥1 symptom: *** 79.3% (95% CI, 13.4–95.1) VE against quantitative culture-confirmed RSV infection with ≥1 symptom: ^†††^ 88.5% (95% CI, 14.8–98.5)

* Data cannot be directly compared as no universal approach was used across trials. Heterogeneity has previously been observed when comparing values detected by luciferase reporter and PRNT assays [67]. ^†^ Confirmed on ≥2 consecutive days with ≥1 clinical symptom of any grade from two categories or ≥1 grade 2 symptom from any category. ^‡^ At least one viral culture result at or above the lower limit of quantitation from Day 2 after challenge to discharge. ^§^ Assessed 6 months after vaccination. ^¶^ RSV-neutralising antibody titres were measured by luciferase reporter gene assay. ** ≥2 measurements above the lower limit of quantification and ≥1 symptom from 2 different categories (upper respiratory, lower respiratory, systemic) from the symptom scorecard or any grade 2 symptom from any category. ^††^ Assessed 14 days after vaccination. ^‡‡^ Trial design included several populations, including older adults. ^§§^ Immunogenicity assessments included IgG ELISA, IgA ELISA, PRNT subtype A, PRNT subtype B, and RSV G-protein-specific (G [A] and G [B]) IgG ELISAs. ^¶¶^ Post-vaccination GMFR were calculated based on the log average of titres measured at Weeks 2, 4, and 6 per participant. *** At least 2 detectable measurements at or above the lower level of detection on ≥2 consecutive days with ≥1 positive clinical symptom of grade 2 or higher from any category in the symptom scoring system (upper respiratory, lower respiratory, systemic). ^†††^ At least 2 detectable measurements at or above the lower level of detection on ≥2 consecutive days with ≥1 positive clinical symptom of any grade from 2 different categories in the symptom scoring system (upper respiratory, lower respiratory, systemic) or 1 grade 2 or higher symptom from any category. ARI, acute respiratory infection; CI, confidence interval; ELISA, enzyme-linked immunosorbent assay; GMFR, geometric mean fold rises; LRTD, lower respiratory tract disease; PRNT, plaque reduction neutralisation test; RSV, respiratory syncytial virus; RT-PCR, reverse transcription polymerase chain reaction.

### 3.2. The G Protein

Targeting the pre-F protein may induce a selective advantage for mutations in multiple antigenic sites; therefore, targeting more than one protein may be a valuable option for new vaccines [54]. Aside from the F protein, the G protein is the only other RSV antigen that induces neutralising antibody responses [22,68,69]. Although the F protein is more conserved across RSV strains than the G protein, the G protein does display a small central conserved domain (CCD) [70]. The CCD is essential for infectivity in vivo, mediates attachment to epithelial cells, and is implicated in alterations of the host immune response that result in airway inflammation. Targeting the RSV G protein in addition to the native F protein may allow for a more comprehensive degree of protection against disease, encompassing antiviral activity, blockade of cellular infection, and promotion of anti-inflammatory activity [70]. This is supported by data from infants and young children aged <2 years diagnosed with acute RSV infection; pre-F antibodies displayed the greatest neutralising activity (55–100%), followed by G antibodies (0–45%). Higher concentrations of each antibody were associated with lower clinical disease severity [71]. Furthermore, data from a preclinical study of a recombinant Modified Vaccinia Ankara (MVA)-based RSV vaccine in two different animal challenge models demonstrated that targeting both the F and G proteins increased vaccine immunogenicity and efficacy to provide more robust protection against RSV than targeting either F or G alone [72]. 

Antibodies against the F protein have been shown to increase with age into adulthood, whereas antibodies against the G protein—specifically those that bind to antigenic site G-4 containing the CCD—decline [71,73]. As the CCD is highly conserved among all RSV strains, low levels of CCD-targeting antibodies may not be simply explained by G protein variants. This could suggest a differential decay of anti-G long-lived plasma cells and/or memory B cells compared with those of anti-F following RSV infection [73]. The age-related decline in CD4^+^ T cell populations and impaired T cell response may be a contributing factor. As CD4^+^ T cells are critical in the affinity maturation of B cells, their declining numbers in older age may hinder the generation of high-affinity neutralising antibodies targeting the G protein [1]. The significant drop in G protein antibodies in adults may contribute to the sustained susceptibility to RSV infections throughout life, highlighting the G protein as a compelling target to complement the F protein for protection against RSV [73]. 

### 3.3. Other Immune Mechanisms of Protection

The immune correlates of protection that contribute to the increased susceptibility to severe RSV illness in older adults have not been fully elucidated. Neutralising antibody levels are inversely correlated with the risk of hospitalisation during acute RSV infection in older adults. However, there is no established threshold of neutralising antibody titres that provides absolute protection in this population [74]. A broad variety of neutralisation assays are currently in use to measure RSV-neutralising antibody titres, [75] and the lack of a standardised measurement approach means that neutralising antibody titre data cannot be directly compared between vaccine candidates. Heterogeneity has been observed when comparing values detected by plaque reduction neutralisation test and luciferase reporter assay [67]. Furthermore, neutralisation assays may not detect all neutralising antibodies for RSV, as they are not optimal for detecting anti-G-neutralising antibodies [67,76]. Additionally, the importance of non-neutralising antibodies that are the cause of enhanced respiratory disease following formalin-inactivated RSV vaccination is understood [77,78]. However, as this review focuses on the pursuit of an RSV vaccine in older adults, original antigenic sin and its links with priming are beyond the scope of this paper. More importantly, neutralising antibody titres may not provide the whole picture for protection. Upon RSV challenge, adults with high concentrations of F and G antibodies still have a 25% risk of being re-infected, highlighting that RSV protection is far from complete and is short-lived [79,80,81]. Data from a human challenge study with the Ad26.RSV.preF vaccine in healthy adults aged 18–50 years demonstrated that RSV infection was still possible in adults with high antibody titres after vaccination [58]. Vaccine candidates targeting the F protein have shown positive CD4^+^ and CD8^+^ T cell responses against RSV in adults aged >18 years (Table 2), which, together with neutralising antibodies, may contribute to their high efficacy against RSV infection. The vaccine candidates GSK3844766A [44,45] and RSVpreF [55] have both shown high vaccine efficacy against RSV infection, yet neutralising antibody titres appear to decline from 6 months after vaccination [37,38,39,56]. Protection against RSV infection has been demonstrated even with low neutralising antibody titres, as shown in a human challenge study with MVA-BN-RSV in healthy adults aged 18–50 years [63,64]. Taken together, these data could signify that neutralising antibodies may not be the only possible mechanism of protection against RSV. Indeed, experimental infection of healthy adults with RSV found that virus-specific nasal IgA was a stronger correlate of RSV protection than serum neutralising antibody titres. However, virus-specific IgA memory B cell responses were absent after RSV infection, which may represent an immune evasion strategy by the virus and partly explain susceptibility to re-infection [80]. This suggests that the ideal RSV vaccine may need to induce durable mucosal antiviral IgA and memory B cells to overcome the impaired local immunity seen after infection [80]. 

To understand why older adults are at risk for severe RSV, an immune profiling study was conducted comparing healthy young adults and older adults. Both groups had comparably high serum RSV-neutralising antibody titres and nasal mucosal IgA levels, suggesting that other immune system deficits are present in older adults that make them more susceptible to severe RSV infection [82]. Significantly reduced RSV-specific memory CD4^+^ and CD8^+^ T cell responses were observed in older adults versus younger adults, suggesting a role for RSV-specific T cell responses in RSV severity [7,21,82]. This is supported by Phase 1/2 data from a pre-F subunit vaccine candidate, which identified a decline in CD4^+^ T cell numbers in older adults compared with healthy adults [39,40]. The CD4^+^ T cell response is required for RSV viral clearance, immunoglobulin class switching to IgA, and differentiation of B cells into long-lived plasma cells and memory B cells [83]. The observed decline in CD4^+^-mediated immunity could increase the susceptibility of older adults to severe RSV disease, demonstrating the need for a vaccine that can boost the population of CD4^+^ T cells to generate long-lived vaccine-induced responses despite immunosenescence [39,83]. 

Older age has also been associated with a lack of RSV-specific interferon-γ (IFN-γ)-producing CD8^+^ T cells, which are essential for viral clearance and provide protection against secondary infection [84]. In vitro RSV-stimulated human peripheral blood mononuclear cells from adults aged >65 years were deficient in RSV-specific IFN-γ production compared with those from adults aged 20–40 years. Restoring the impaired cellular response to natural infection in older adults may be a consideration for vaccines targeting this population [82]. In healthy adults aged 18–55 years, peptides from the M, N, P, NS1, and NS2 RSV internal proteins were all shown to induce IFN-γ production [6], which was replicated in Phase 1 and 2 clinical trials of a vaccine candidate targeting these internal proteins in addition to the F and G proteins [61,62].
vaccines-11-00382-t002_Table 2Table 2Cellular immune responses to current Phase 3 RSV vaccine candidates in healthy and older adults in clinical development.Vaccine Candidate, SponsorFormulation,AntigenClinical Trial PhasePopulationsAssessedNRSV-Specific IFN-γ-Secreting T Cells GMFRs vs. Baseline *RSV-Specific IFN-γ-Secreting T Cells GMFRs ≥1-Year Post-Vaccination *RSVpreF,Pfizer [38]Bivalent subunit,Prefusion FPhase 1/2Older adults aged 65–85 years317(n = 64)Month 1: T cell responses declined but remained above baseline–GSK3844766A,GSK [39,56]Recombinantsubunit,Prefusion F +AS01_E_ adjuvantPhase 1/2Adults aged18–40 years48Day 31: CD4^+^ 2.2–3.2–Phase 1/2Older adults aged 60–80 years1005Day 31: CD4^+^ 1.7–3.2–Phase 3Older adults aged ≥60 years1653 (n = 566)Day 31: CD4^+^ increased from 191 to 1339 events/10^6^ PBMC; CD8^+^ no detectable responseMonth 6: CD4^+^ declined to 666 events/10^6^ PBMC; CD8^+^ no detectable response–Ad26.RSV.preF, Janssen [41,57,59]Adenoviral vector 26,Prefusion FPhase 1Older adults aged ≥60 years72Day 28: ≥2.1 ^†^≥2.3-fold 28 days after second vaccination at 1 year ^†^; maintained above baseline 2 years after vaccination ^†^Adeno 26 viralvector Prefusion F + Prefusion F subunitPhase 2bOlder adults aged ≥65 years5782 (n = 195)Day 14:increased from 34 to 444 SFC/10^6^ PBMC^†^1.5 years: 143 SFC/10^6^ PBMCMVA-BN-RSV,Bavarian Nordic [61,62]Bivalentvector-based,RSV F, G (A), G (B), M2-1 and NPhase 1Adults aged18–49 years631.8–3.7 ^‡^–Older adults aged 50–65 years2.4–3.8 ^‡^–Phase 2Older adults aged ≥55 years420(n = 113)Week 1: ^§^ 5.4–9.7Week 5: ^¶^ 2.4–4.7Maintained above baseline 1 year after initial vaccination* Data cannot be directly compared as no universal approach was used across trials. ^†^ Median frequency of F-specific IFN-γ-secreting T cells. ^‡^ Calculated based on GMSFUs for 5 peptide pools measured at Weeks 1, 2, and 5 per participant. ^§^ One week after the initial vaccination. ^¶^ One week after the second vaccination. GMFR, geometric mean fold rises; GMSFU, geometric mean spot forming unit; IFN-γ, interferon-γ; PBMC, peripheral blood mononuclear cells; RSV, respiratory syncytial virus; SFC, spot-forming cells.


## 4. MVA-BN-RSV: A Late-Stage Vaccine Candidate Targeting Multiple RSV Antigens

The Bavarian Nordic MVA-based RSV vaccine is the only late-stage bivalent vaccine candidate that targets five proteins, including the F, G (A), and G (B) proteins in addition to the highly conserved internal M2-1 and N proteins, to specifically target the cellular immune response (Figure 2). MVA-BN-RSV is currently in Phase 3 trials (NCT05238025). It has been given breakthrough therapy designation by the FDA and granted access to the priority medicines (PRIME) scheme by the EMA in active immunisation for the prevention of lower respiratory tract disease caused by RSV in adults aged ≥60 years [85].

The advantage of a multi-component vaccine was demonstrated in preclinical mouse models upon intranasal RSV challenge after vaccination with MVA-BN-RSV. The study revealed that complete protection against RSV challenge was dependent on CD4^+^ and CD8^+^ T cells as well as RSV-specific antibodies, including mucosal IgA. These findings indicated that vaccination with MVA-BN-RSV induced a broad adaptive immune response and suggest that protection against RSV might not be correlated to a single immune response [86].

In clinical trials, the MVA-BN-RSV vaccine candidate has been shown to be well tolerated and immunogenic. A Phase 1 study demonstrated that MVA-BN-RSV elicited both T cell and antibody immune responses, with no differences in safety and immunogenicity observed between adults aged 18–49 years and 50–65 years [61]. A Phase 2 trial confirmed that vaccination with MVA-BN-RSV stimulated broad T cell responses across all five encoded RSV antigens (more than 5.4-fold from baseline) coupled with broad humoral responses in adults aged ≥55 years [62]. The broad immune response with MVA-BN-RSV may indicate activation of adaptive immune responses against RSV, which could contribute to different pathways of protection [61,62]. In addition, as discussed earlier, the vaccine candidate may offer a lower risk of mutational escape variants compared with vaccines targeting a single antigen and re-establish high-affinity neutralising antibodies targeting the G protein [1]. A single dose was also shown to increase levels of neutralising antibodies IgG and IgA (1.6–3.4-fold from baseline at Day 14) and they remained above baseline for 6 months. However, a dose at 12 months elicited a booster effect in humoral and cellular responses up to 2.8-fold from pre-boost levels [62]. Compared with other vaccine candidates, Phase 3 data showed a 10.5-fold rise from baseline in RSV A-neutralising antibodies following vaccination with RSVPreF3 at Day 31, yet this dropped to 4.4 after 6 months [56]. Similarly, Phase 2b data for Ad26.RSV.preF + preF subunit showed a 13.5-fold increase in RSV-neutralising antibodies at Day 14 following vaccination, which reduced to 2.8 at 1 year but still remained above baseline [57,59].

Despite differences in neutralising antibody titres and different mechanisms of action, RSV vaccine candidates have shown similar vaccine efficacy in human challenge trials. MVA-BN-RSV was recently investigated in a human challenge trial to assess the efficacy of vaccination in reducing RSV viral loads compared to a placebo in healthy adults aged 18–50 years. A significant reduction in viral load was observed in the vaccine group compared with the placebo group (*p* = 0.017). Additionally, vaccinated subjects showed a significant reduction in clinical symptoms typically associated with RSV infection, demonstrating a vaccine efficacy of 79–89% for preventing symptomatic RSV infection [63,64,65,66]. Vaccine efficacies of 86.7–100% and 51.9–55.1% (conservative definition) have been reported for RSVpreF [55] and Ad26.RSV.preF [58], respectively, for preventing symptomatic infection in healthy adults. However, it should be noted that human challenge studies are designed to evaluate efficacy in the prevention of infection rather than the prevention of severe disease.

## 5. Other Vaccines in Early Development

We are at an exciting time in RSV vaccine development for older adults, with expanding technologies and several candidates currently in early-stage clinical trials in addition to those discussed in late-stage clinical trials [87]. Three RSV G-protein-based vaccines are in clinical development, including Advaccine’s BARS13/ADV110, which contains recombinant RSV G protein and cyclosporine A, and has entered Phase 2 clinical testing in older adults aged 60–80 years after demonstrating immunogenicity and safety in Phase 1 [87,88,89]. Meanwhile, a candidate targeting the SH protein of RSV A demonstrated high immunogenicity with sustained antigen-specific antibody responses alongside an acceptable safety profile in Phase 1 testing of adults aged 50–64 years [90]. These early-phase candidates highlight that multiple approaches are being undertaken in RSV vaccine development for older adults beyond targeting the F protein.

## 6. Conclusions

RSV remains a major health concern among infants and older adults, with an increased risk of severe symptom development and death among these groups. Although immunoprophylaxis is available for some infants, there is no preventative RSV vaccine available for all infants or older adults. Currently, there are five RSV vaccine candidates in late-stage development for older adults: four targeting only the pre-F protein and one targeting the F, G (A), G (B), M2-1, and N proteins. Vaccine candidates have demonstrated robust neutralising IgG and cellular responses and have shown efficacy for protection against RSV infection in clinical trial settings. Three pre-F candidates in Phase 3 clinical trials have announced positive top-line data, revealing significant and clinically meaningful efficacy in adults aged ≥60 years. Until the long-term effectiveness of these candidates is confirmed, it is important that research continues to investigate the RSV immune response. Efforts are underway to further understand how the choice of antigenic target site may impact vaccine effectiveness and whether targeting single or multiple antigens will be the most effective public health strategy for protecting against RSV. Ultimately, this will help improve our understanding of how best to exploit its mechanisms and achieve effective protection against the virus. Overall, we are at an exciting time of vaccine development with several candidates currently in clinical trials; however, the quest for a successful RSV vaccine is not over yet.

## Figures and Tables

**Figure 1 vaccines-11-00382-f001:**
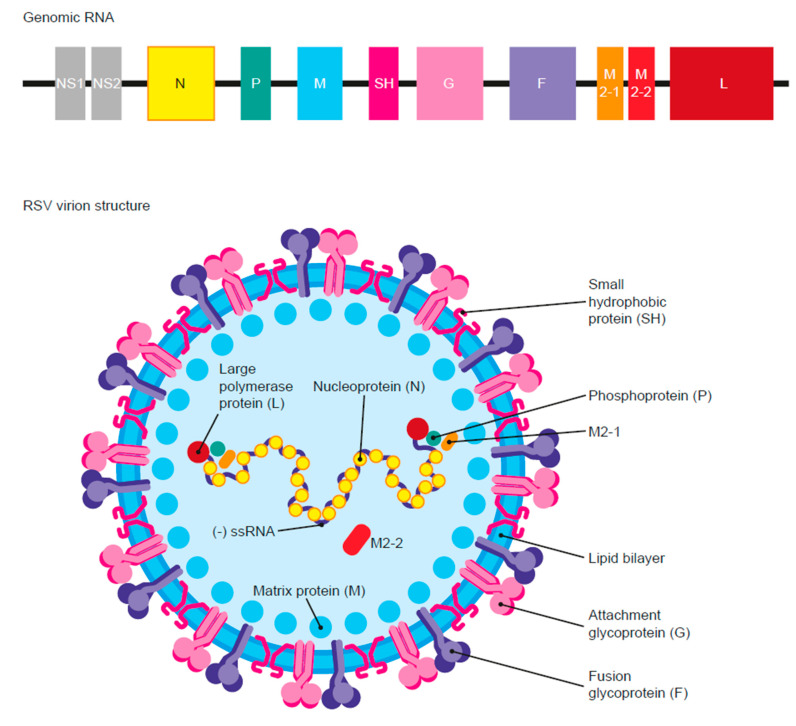
Respiratory syncytial virus (RSV) genome and virion structure. RSV is a negative-sense, single-stranded RNA virus encoding 11 proteins: nine structural and two non-structural proteins. The fusion glycoprotein (F), attachment glycoprotein (G), and the small hydrophobic protein (SH) are transmembrane proteins comprising the viral envelope. Matrix proteins (M) are found at the inner side of the viral envelope. Four nucleocapsid and regulatory proteins function as viral transcription factors: nucleoprotein (N), phosphoprotein (P), large polymerase protein (L), and M2-1 and M2-2 proteins. NS1 and NS2 are the two non-structural proteins. Adapted from ref. [18] CC BY 4.0.

**Figure 2 vaccines-11-00382-f002:**
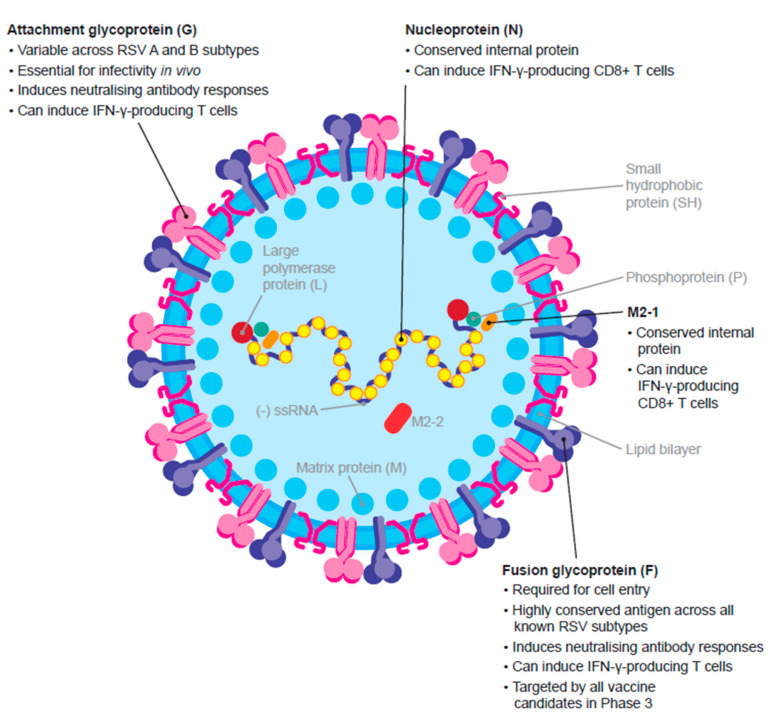
RSV virion structure and antigen targets for the MVA-BNA-RSV vaccine candidate. The vaccine incorporates five different RSV antigens (F, G (A), G (B), M2-1, and N) to stimulate a broad immune response against both RSV subtypes (A and B). IFN- γ, interferon-γ; RSV, respiratory syncytial virus. Adapted from ref. [18] CC BY 4.0.

## Data Availability

No new data were created or analysed. Data sharing is not applicable to this article.

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
