# Peer review of "The Quest for a Respiratory Syncytial Virus Vaccine for Older Adults: Thinking beyond the F Protein"

_vaccines, 2023, doi:10.3390/vaccines11020382_

Round 1

Reviewer 1 Report

This is a review of the manuscript #2114147 submitted to Vaccines by Jenkins et. al. In this manuscript, the authors summarize the status of RSV vaccine candidates as of date. The article begins with a clear and succinct overview of RSV, clinical impact pathogenesis and then delves into the molecular features of the virus that are targeted for vaccine development. The article is written in a clear understandable manner and graphics show will help someone new to the field. Following are the salient points that in my opinion could enhance the quality of the manuscript.

1.     In addition to poorly understood immune response to RSV infection, there should be some discussion of RSV latency / persistence in extra-pulmonary tissues and how that contributes to RSV resurgence during immune senescence. While the focus of the manuscript is about RSV vaccines, this is an important contributor to RSV in the elderly.

2.     The authors should discuss why a significant proportion of anti-RSV F antibodies fail to neutralize the virus and further why neutralization does not necessarily reduce disease severity. Please discuss original antigenic sin in the context of RSV immune response.

3.     While the authors do cover most RSV vaccine candidates they do miss discussing approaches such as codon-optimized, codon-deoptimized vaccines and live attenuated vaccine candidates that are currently under development. Including these in the tables listed would again improve the breadth of the review.

Reviewer 2 Report

Figure Labelling in both figure 1 and 2 is incorrect. ‘Fusion protein’ is pointing to the tetrameric protein and ‘Glycoprotein’ is pointing to the trimeric protein.

Line 176-193: The line of reasoning that vaccination against F alone would result in antibody escape mutant viruses is speculative. Presumably this pressure has already existed for decades as infection with RSV generates already antibodies against such epitopes and as stated there is very little divergence between even subtype A and subtype B. It could instead be argued that increasing neutralising antibodies by 10fold in individuals through widespread vaccination with preF would instead decrease RSV replication / circulation and therefore make it less likely for mutant viruses to be produced in vaccinated individuals than unvaccinated individuals. Evidence that a single monoclonal doesn’t neutralise all strains is weak.

Line 224-228. Does reference 64 refer to preF + G or non-stabilized native F + G? if it is the later then addition of G may provide no additional advantage to optimal preF only to inferior native F.

Section 4 is heavily biased in favour of the vaccine being produced by Bavarian Nordic (Company to which the authors are affiliated). This section should better reference against results found by other vaccine developers such as Pfizer and GSK. For example, Line 319-321: It is mentioned that neutralising antibodies increase by 1.6-3.4 fold above base line following vaccination. It needs to be noted that this is well below the 10fold rise in neutralising titres reported by comparator vaccines. Also, Line 330-332, other vaccine developers have also reported results from human challenge studies and these should be included for comparison.

Reviewer 3 Report

Overall, the review gives a good overview of current RSV vaccine approaches that are in late-stage clinical trials targeting the older adult population. It gives a comprehensive concise background of the RSV disease burden, RSV and antigen targets for vaccines. It highlights the currently late-stage vaccines that are developed in Europe and the USA, mainly focusing on their efficacy and highlighting their immunogenicity.

At times the review is a bit unclear on target population it focuses on as the infant/young children population is taken along, while the focus is clearly on the older adults. This should be reflected better in the title (add older adults) and in the abstract as for instance in line 21 “in this population” is used that is referring to older adults I assume.

Major comments:

A)      In lines 176-192 the authors stipulate that a single antigen target is likely to lead to immune escape and use the failed monoclonal antibody suptavumab as an example. While it is true that this antibody failed due to antigen changes leading to loss of efficacy this is highly unlikely for a polyclonal immune response induced by a pre-F based vaccine targeting the full antigen. Here it is unlikely that single changes will lead to escape and it has been shown by the GSK vaccine that efficacy against RSV B is high, while the vaccine is based on RSV A. Please revise the section in this respect.

B)      The major mode of action of protein subunit vaccines based on RSV pre-F that have shown now efficacy in phase 3 studies is likely relying on neutralizing antibodies as these are subunit vaccines and as post-F vaccines have failed. It is therefore needed to also describe in more depth current state of the art immunological assays for detection of the response for this F antigen, but also G. This will help to put the data in perspective and to be able to position other vaccines in development.

C)      I miss a section at the end of the review on other vaccines in earlier stages of development that may also incorporate other antigens than RSV preF

Minor comments:

A)      Section 4 is too long in comparison to the 4 vaccines that are based on preF and from which two have phase 3 efficacy data made public. This should be shortened or more balanced. Figure 2 in this section should be added more to figure 1.

B)      Lines 26-35 of the introduction only reports data from UK and USA, rest of world impact is missing although disease burden should is equally relevant.

C)      Line 60: definition of RSV infection and re-challenge is unclear. If adaptive immunity is established after first RSV infection it can only prevent from re-challenge, not RSV infection as that would be first encounter with the virus. Please clarify what is meant.

D)      Line 65: most individuals have RSV humoral immunity after the first virus encounter. That immunity is measurable also in older individuals, so it seems durable – the levels may be dropping though, so it may be the durability of a high enough magnitude to prevent reinfection

E)      Lines 100-112: a comparison is made between monoclonal Ab protection from RSV infection in infants and adults. To be able to appreciate the statements, please provide dose/BW used in these studies. Similarly, please give a dose level for Nirsevimab in the next paragraph.

F)       Line 126: 78% protection should not be considered a modest level of protection, especially as SOC so far was much lower.

G)     Line 151: please expand on the failed attempts to develop an RSV vaccine as these learnings were valuable for current vaccine efforts

H)     Line 153-175: please give some more details on the vaccines, while that is in table 1, the text would benefit for easy reference

I)        Table 1: in the neutralizing Ab GMFR section IgG is mentioned, please use a separate column to report binding Ab titers for clarity. For the MVA vaccine a challenge trial is described in the text, but omitted from the table. Please add. For the phase 2a of the MVA vaccine, 61 subject were enrolled. How can efficacy be reported with such low numbers and where efficacy endpoints were likely not a primary outcome, but a posthoc analysis. Please indicate how this data was generated.

J)        Lines 229-241: If CCD is conserved and inducing Ab why would it be a good target for vaccines? It is seen by the immune system with each reexposure, but seems not to induce protective immunity. Please discuss this.

K)      Section 3.3: It seems to disregard the high efficacy reported for the subunit vaccines. While IgA and T cells likely contribute in natural infection, this is not necessarily the case for vaccine induced protection if Ab levels are high enough

L)       . Please make clearer the difference of the induced immunity and the need for these arms of immunity.

M)    Table 2: can you use the same way of reporting of GMFR throughout?

N)     In the future directions section I miss the mentioning of apparently successful vaccine phase 3 outcomes and what that means for future vaccine approaches. Please include.
